# OpenReview forum: "LLM-based Multi-Agents System Attack via Continuous Optimization with Discrete Efficient Search"
_colmweb.org/COLM/2025/Conference — COLM 2025_

### Official Review · Reviewer_9qgi · 2025-05-12

**Rating:** 5
**Confidence:** 3
**Ethics Flag:** 1

**Summary:**

This paper introduces CODES, a novel adversarial attack framework targeting Large Language Model (LLM)-based Multi-Agent Systems (MAS). The method combines continuous optimization with discrete efficient search to create self-replicating adversarial prompts, achieving jailbreaks through a single intervention in realistic multi-agent settings. The paper is well-motivated and addresses a timely and important problem. It proposes three practical attack scenarios and demonstrates the effectiveness of the approach with comprehensive experiments, including evaluations across different LLMs and settings. However, the paper could benefit from improved clarity in algorithm explanations, additional real-world grounding, and stronger evaluations on black-box models.

**Questions To Authors:**

- Can you provide a concrete example walk-through for Algorithm 1 and Algorithm 2 to better clarify the optimization process?

- How would the inclusion of RAG or tool-augmented agents affect the attack's propagation or success rate?

- Can the attack be extended or adapted to multimodal agents or agents with strong task-specific constraints?

- How does your method perform under different MAS topologies or communication schemes beyond randomized pairing?

- Are there any countermeasures or alignment techniques you believe would be particularly effective against CODES?

**Reasons To Accept:**

- The topic is timely and relevant given the increasing deployment of LLM-based MAS in real-world applications.

- The proposed attack method, CODES, is novel in combining continuous optimization with discrete search in the MAS jailbreak setting.

- Experimental results are strong and include comprehensive ablation studies.

- The formulation of different attack scenarios is thoughtful and well-structured.

- The approach demonstrates transferability to black-box models, indicating practical relevance.

**Reasons To Reject:**

- Some algorithmic details (e.g., Algorithm 1 and 2) are dense and would benefit from illustrative examples to enhance clarity.

- The evaluation on black-box models, while promising, is relatively shallow. More rigorous and broader testing is needed, especially since black-box settings are common in real-world deployments.

- The discussion of how the proposed attack method relates to real-world MAS applications could be expanded.

- It remains unclear how external factors, such as retrieval-augmented generation (RAG) or tool usage, might affect the attack's success or transferability.

- The paper might benefit from additional attack scenarios or case studies that are more grounded in application-specific contexts.

---

> ### Author Response · Authors · 2025-05-30
> **Response to Reviewer 9qgi 1/2**
>
> Question1 and Weakness 1 will be in another thread. Thanks for all the comments and questions.
>
> **Response to Weakness 2: *More testing in black-box.***
>
> The method is optimization-based, so we mainly evaluate on the white box settings, as other optimization based evaluation as in GCG and ADC.
>
> **Response to Weakness 3: *Real-world MAS applications.***
>
> For example, in the future, the vehicles can be connected to each other in a transportation system, and each with a VLM inside. They can communicate with each other. Then, a single interruption in one vehicle can lead to all vehicles do the same, harmful and precise outputs.
>
> **Response to Weakness 4 and Question 2: *How RAG affects the attack.***
>
> If the agent can retrieve outside data from a database, then the attack interface becomes more vulnerable than the current settings since the harmful input can be long and can hide in retrieved documents. But how to retrieve back these harmful documents retains another open research problem. If the retrieved document is long and harmless, then it may decrease the attack success rate. Overall, this is another task and investigating this problem can stand alone as another interesting work.
>
> **Response to Weakness 5: *Application-specific contexts.***
>
> The proposed method can generalize to different datasets and different contexts. Once you optimize on a certain domain-specific train set, it can generalize to unseen data following the same distribution.
>
> **Responese to Question 3: *Extended to multimodal agents or task-specific? ***
>
> The proposed method mainly addresses the text token optimization problem, which is much harder than adversarial image optimization. If, in the multimodal agent setting, the attacker can input adversarial images, then the settings are much easier than the settings in this paper. Previous multiple adversarial methods in image domain can be used.
>
> As for task-specific, if it is all in text, and the context are not too long, then the proposed method can be well extended to these scenarios.
>
> **Responeses to Question 4:  *Different MAS topologies***
>
> If the topology (the order of the agent communication) is predefined and known to the attackers, then the case become easier and attack success rate (ASR) should increase. Otherwise, if the topology is not known, and if some agents communicate less frequently with others, then ASR may decrease until these agents are successfully infected.
>
> **Responeses to Question 5: *Countermeasures or alignment techniques***
>
> For harmful content, using a content filter can reduce ASR. However, a filter usually suffers from over refusal issues, which may influence the performance of completing regular tasks.
>
> For  stealthy behavior such as saving a confidencial file, then training another agent for identifying unsafe code can be helpful.

---

> > ### Comment · Reviewer_9qgi · 2025-06-09
> > **Response to Authors**
> >
> > Thank you for your response and additional clarifications. While the work is interesting and timely, my concerns remain regarding black-box robustness and real-world applicability. I will maintain my score but welcome further discussion.

---

> ### Author Response · Authors · 2025-06-01
> **Response to Reviewer 9qgi 2/2**
>
> **Response to Weakness1 and Question 1: *A concrete example walk-through for Algorithm 1 and Algorithm 2 to better clarify the optimization process?***
>
> Let $z_1, z_2, …, z_n$ denote the embedding vector to be optimized. For all i, $z_i$ is a dense $d$-dimensional vector where $d$ is the embedding size of the LLM. After optimization, we use the arg max index of each embedding vector for the optimized token.
>
> At the optimization step $t$, we first compute the gradient and then the momentum buffer of all  embedding vectors ${z_i}_{i=1}^n$.
>
> Algorithm 1 described how to update ${z_i}_{i=1}^n$ according to the momentum buffer. The overall idea is to update only one index of one embedding vector (and then project the embedding vector to the probability space again). The question is to choose which index of which embedding vector. The answer from Algorithm 1 is to try some candidates and choose the candidate with best loss:
>
>
> **Generate one candidate**: we uniformly sample j from [1, 2, …, n]. Thus for this candidate, we choose the j-th embedding vector. And then we choose an index, say k,  from the j-th embedding vector, among the indexes with top-K large gradient:
>                [index_i, index_2, …, index_K] = Top_k(gradient(z_j)[i])
>                k ~ Uniform ([index_i, index_2, …, index_K])
> Thus for this candidate, we only update the k-th index of the j-th embedding vector using the momentum buffer.
>
> **Choose the candidate with the best loss**. For each candidate, we know how the embedding vectors are updated, and can compute the optimization loss after update. We choose the candidate with the lowest updated loss as the final update of this step because it will go to a lower loss.
>
> **Convert the updated embedding vector into target sparsity**. After each update step, the sparsity of the embedding vector may increase. However, we want a very sparse embedding vector in the end. At step $t$, we follow ADC to convert the updated embedding vector into a pre-computed sparsity. We use the same convert algorithm as in ADC.
>
> **Evaluation**. As we mentioned, we use the arg max index of each embedding vector for the optimized token. Then at each step, we evaluate if the (discrete) language token (from  the arg max index of each embedding vector) can jailbreak or not. If yes, the optimization is stopped. If not, the optimization will continue until all $T$ optimization steps are executed.

---

### Official Review · Reviewer_3nxL · 2025-05-13

**Rating:** 4
**Confidence:** 3
**Ethics Flag:** 1

**Summary:**

This paper proposes CODES, a token-level adversarial attack designed to compromise LLM-based multi-agent systems (MAS) through a single interaction with one agent. The attack leverages continuous optimization with discrete search to generate adversarial suffixes that propagate across agent communications. The authors evaluate their method under three attack settings and demonstrate high success rates across various open-source models, as well as some transferability to black-box models like GPT-4o.

**Reasons To Accept:**

1. The problem setting is timely and realistic.
2. Experiments across different LLMs show good ASR attack and partial transferability to GPT-4o, outperforming prior jailbreak methods (e.g., GCG, ADC).

**Reasons To Reject:**

1. The writing is weak throughout—starting with a poorly structured abstract (the abstract is not logically fluent) and extending into a dense, jargon-heavy method section that lacks intuitive explanation.

2. Lacks novelty and did not compare with proper baseline. The proposed method is a modest refinement over existing token-level optimization methods, and the broader threat model in MAS has already been explored in recent works (see below). These works already investigate propagation attacks, adversarial coordination, and structured jailbreaks in MAS. Overall, while the paper identifies an important threat vector, the contribution is incremental and the execution lacks the polish and clarity needed for publication.

Refs:
Red-Teaming LLM Multi-Agent Systems via Communication Attacks
Combating Adversarial Attacks with Multi-Agent Debate
MultiAgent Collaboration Attack: Investigating Adversarial Attacks in Large Language Model Collaborations
Amplified Vulnerabilities: Structured Jailbreak Attacks on LLM-based Multi-Agent Debate papers

---

> ### Author Response · Authors · 2025-05-30
> **Response to Reviewer 3nxL**
>
> Thanks for the time and reviews.
>
> **Response to Weakness 1: *Intuitive explanation***
>
> **Overall intuition**: the method optimizes a harmful, repetitive suffix so that input (harmful words + suffix) produces output (harmful words + suffix), enabling precise harmful content propagation across agent networks.
>
> Intuition for the optimization: **CODES keep a continuous optimizable variable z, and search for discrete candidates for evaluation**.
>
> Compared with optimization in discrete space, it’s more training stable. (GCG optimizes all tokens in discrete), ADC, GDBA and other projected gradient descent (PGD)-based methods project from discrete to continuous per step, bouncing around. CODES always optimize in continuous (as in line 129 and line 137).
>
> More generalizable. Compared to PGD, CODES can maintain a large batch size by search (PGD can only do batchsize=1) per initialization. Thus, CODES can better escape from a local minimum, reach a lower loss. (as in line 127 and line 141).
>
> **Response to Weakness 2: *lacks comparison with proper baseline and lacks novelty***
>
> Thanks for mentioning these related works [1,2,3,4], we will add them properly in the reference.
>
> [1] Red-Teaming LLM Multi-Agent Systems via Communication Attacks
>
> [2] Combating Adversarial Attacks with Multi-Agent Debate
>
> [3] MultiAgent Collaboration Attack: Investigating Adversarial Attacks in Large Language Model Collaborations
>
> [4] Amplified Vulnerabilities: Structured Jailbreak Attacks on LLM-based Multi-Agent Debate
>
> But we have **major differences with all these papers.**
>
> **Briefly, all these attacks are not a formal mathematical optimization method, but rather a heuristic feedback-based approach.** They can not precisely control the output, making their attacks less harmful and capable.
>
> In detail:
>
> 1) **Threat model setting.** [1,2,3,4] all based on debate. They require a fully controlled harmful agent, and require this agent to interact with other agents multiple times. But we only need to interact with the system once. We do not need a fully controlled agent. Our setting is harder and more practical.
> 2) **Method.** [1,2,3,4] are all based on prompting. Ours are based on optimization on tokens. Their methods are more like template-based methods as we discussed in line 43-52. For example, [1,2] use agents debate to reduce prompt harmfulness, [3,4] both involves ‘Iterative Refinement’, which is to prompt an llm to rewrite the harmful prompt to be less obvious.
>
>
> **Response to Weakness 2: *just an optimization-based method and lacks novelty.***
>
> **Previous optimization-based methods don’t work in our difficult agent setting** (ASR=0 in tab.1).
>
> CODES makes improvements on 1) Improved algorithm, combining search with optimization. 2) Improved objective: implicit weighted objective. 3) Improved ensemble: scale, order and diversity.

---

> > ### Comment · Reviewer_3nxL · 2025-06-09
> >
> > I thank the authors for their response. While I appreciate the clarification, my concern regarding the lack of comparison with other multi-agent baselines [1][2][3][4] still remains.
> >
> > Moreover, given that these prior works propose similar settings, the novelty of the paper specifically the part highlighted by Reviewer 1, that “the paper proposes three well-defined and realistic MAS attack scenarios, grounded in how real-world LLM-based multi-agent systems operate with limited user access”, no longer seems as compelling. A more thorough discussion of related work is crucial to properly situate and define the paper's contribution.
> >
> > Even if the baselines are template-based, including them would help demonstrate the performance gains achieved by the proposed method.
> >
> > Additionally, as mentioned earlier, the paper would benefit from further proofreading to eliminate unclear logic—particularly in the abstract.
> >
> > Therefore, I will be keeping my original scores, but welcome further discussion.
> >
> > [1] Red-Teaming LLM Multi-Agent Systems via Communication Attacks
> >
> > [2] Combating Adversarial Attacks with Multi-Agent Debate
> >
> > [3] MultiAgent Collaboration Attack: Investigating Adversarial Attacks in Large Language Model Collaborations
> >
> > [4] Amplified Vulnerabilities: Structured Jailbreak Attacks on LLM-based Multi-Agent Debate

---

> > ### Comment · Reviewer_3nxL · 2025-06-09
> >
> > Regarding comparisons with other baselines, I understand and appreciate the point that "other attacks are not formal mathematical optimization methods (also I am not sure whether we should call GCG or ADC or CODES formal mathematical optimization methods, I think white-box or gradient-based approaches might be better), but rather heuristic feedback-based approaches." That said, this seems to reflect a broader distinction between white-box and black-box attack settings, where having access to model gradients naturally makes targeted attacks more tractable. However, given the existence of several effective black-box multi-agent jailbreak attacks, I remain curious about how the proposed method would perform when evaluated against simpler black-box multi-agent attack strategies.
> >
> > In addition, while your approach is white-box and appears to require only one interaction round, it still relies on gradient-based optimization in continuous space, followed by discrete search in the word embedding space (similar to methods like GCG or ADC). Given this, I am not fully convinced by the reasoning for omitting comparisons with those other baselines. If we allow rewriting using structured multi-agent attacks or adopt their universal jailbreak template (e.g., https://arxiv.org/pdf/2504.16489), it would be helpful to see how your method performs in comparison.

---

> > ### Author Response · Authors · 2025-06-10
> > **Reply to Official Comment by Reviewer 3nxL thread 1**
> >
> > Thanks for your reply.
> > Your main concern is about the difference of our setting and the four papers you mentioned [1,2,3,4].
> >
> > **We have main differences in settings compared with [1,2,3,4]**.
> >
> > 1, All of [1234] are based on debate; they require **full control of one agent** or **multiple turns** of intervention. These requirements are less practical. Ours only needs to **intervene once**.
> >
> > 2, **[1,2,3,4] will result in an ASR=0 in our setting**. The settings in [1,2,3,4] do not require precise control of the target agent. In our settings, scenario 1 and scenario 2, we require **an exact match** in our attack goal. Ours posesa  more harmful impact in practical usage.

---

> > ### Author Response · Authors · 2025-06-10
> > **Reply to Official Comment by Reviewer 3nxL 2**
> >
> > Thanks for your reply. Your main concern is about why not compare with baseline in [4].
> >
> > The reason is, our attack scenario requires an exact match. [4], as a prompt-based method (including rewriting, template-based), **result in ASR=0 here.**
> >
> > To be more clear, for example, **baseline [4] can not let all agents repeat a target harmful string *t* by rewriting or by setting a template. [4] can not let a *j*-th agent output an exact string such as f.save(confidential_file) by rewriting or by a template, for any *j***. But the proposed method can succeed in these scenarios.

---

> ### Comment · Reviewer_3nxL · 2025-06-10
>
> Thank you for your response. I agree with your point that white box attacks, particularly those designed for exact matches, are naturally constrained to targeted outputs. For example, in your setup, repeated occurrences of "damn it" are counted as successful attacks.
>
> However, in the broader context of jailbreak attacks, attack success is often defined more loosely. Rather than relying on specific target tokens like "sure, xxx," many evaluations consider any affirmative response that does not include a clear refusal as a success. I don’t believe exact match is the most appropriate criterion for assessing jailbreak behavior. I also question the utility of applying this criterion to HatefulBench, where the goal is to determine whether the model reproduces specific hateful phrases. In jailbreak settings, we should not dismiss alternative approaches simply because they do not produce the exact target output that a given algorithm is optimized for.
>
> While I appreciate the strengths of your approach, my concerns remain. I believe Reviewer 4 articulates my perspective more accurately, particularly regarding black box robustness and real-world applicability. My concerns about using exact match for measuring attack success are closely tied to the exclusive focus on HatefulBench, rather than more widely used benchmarks like AdvBench.
>
> I do appreciate that the paper proposes new MAS setups across three scenarios, tests them on 100 target phrases drawn from the proposed HatefulBench, and introduces CODES (an improved version of ADC) using exact match for evaluation. However, I feel that each of these contributions would benefit from being more clearly explained and evaluated independently.
>
> **For example, if the main contribution is that your approach improves upon ADC, it would be more convincing to evaluate it on AdvBench or HarmBench as ADC does (or use AdvBench or HarmBech with 3 MAS senarios). If the contribution is HatefulBench itself, then a direct comparison with other harmful response datasets would be more appropriate. Similarly, if the key contribution is the three MAS scenarios, it would help to clarify their advantages and usefulness in comparison to prior MAS attack papers.**
>
> As it stands, the paper appears to pursue multiple directions at once, but each thread feels not fully justified with experiments  to me.

---

> > ### Author Response · Authors · 2025-06-10
> > **New Reply to Official Comment by Reviewer 3nxL**
> >
> > Thanks for your reply. We appreciate it that the concern in previous threads about baselines with [1,2,3,4] is addressed. The new concerns are:
> >
> > Reply to Comment in paragraph 2: I don’t believe exact match is the most appropriate criterion for assessing jailbreak behavior.
> >
> > Usually, different jailbreaks use different metrics. For example, if it’s ‘how to make a bomb’ jailbreak, then an exact match is not good. However, **the exact match is a good metric in the proposed jailbreaks**.  Scenario 1, the attack goal is all agents repeat a target harmful string t. Scenario 2, the attack goal is a *j*-th agent output an exact string such as f.save(confidential_file) by rewriting or by a template, for any j. In these cases, an exact match is appropriate.
> >
> >
> > Reply to Comment: if the main contribution is that your approach improves upon ADC,... If the contribution is HatefulBench itself, …’
> >
> > Our contribution is,
> >
> > 1) We propose three novel and practical attack scenarios that allow only a single intervention on one agent from the MAS. (abstract line 4-5)
> >
> > 2) Previous methods can not succeed in these scenarios so we develop CODES that ontinuous-space optimization with discrete-space search. (abstract line 5-7)
> >
> > Reply to Comment: advantages of the three scenarios and how different from previous works.
> >
> > Previous attacks need full control / multiple intervention, and they don’t require precise control. (introduction line 24-27, with example in line 37-47). The reason why we require precise control is that these attacks are more harmful.

---

> > > ### Comment · Reviewer_3nxL · 2025-06-10
> > >
> > > Thanks for the response. My concerns remain, and I will keep my score.
> > >
> > > The paper proposes three MAS scenarios, evaluates them on 100 author-defined hateful targets, and introduces CODES (an improved ADC variant) using exact match. However, the rebuttal does not fully address key issues:
> > > - If the main contribution is improving ADC, evaluation on AdvBench or HarmBench—as ADC does—would be more convincing.
> > > - If the focus is HatefulBench, comparison with existing harmful response datasets is needed.
> > > - If the novelty lies in the MAS scenarios, their advantages over prior MAS attack setups should be clearly articulated.

---

> > > > ### Author Response · Authors · 2025-06-10
> > > > **Reply to Official Comment by Reviewer 3nxL**
> > > >
> > > > Thanks again for the comment.
> > > >
> > > > Our contribution is closer to the third one, 'the MAS scenarios'. In addition, these scenarios are hard to successfully attack, and we also provide the corresponding analysis of why the previous method fails and how we improve based on the analysis.
> > > >
> > > > Response to ‘Advantages over prior MAS attack setups.’
> > > >
> > > > Previous attacks need full control / multiple interventions, and they don’t require precise control. (introduction line 24-27, with example in line 37-47). The reason we require precise control is that these attacks are more harmful.

---

### Official Review · Reviewer_6z7R · 2025-06-01

**Rating:** 7
**Confidence:** 3
**Ethics Flag:** 1

**Summary:**

The authors present CODES with the goal of attacking multi-agent systems with interaction with just one agent. The paper shows a higher success rate and quicker convergence across several open-source LLMs and partial transfer to GPT models. The project also includes the release of HatefulBench, a dataset aimed at testing the propagation of hateful phrases in agent networks.

**Questions To Authors:**

Are there statistical robustness checks? (eg. means +/- SD over several seeds, or a bootstrapped confidence interval over the sampling distribution)

How realistic are the choices for evaluation settings? For example, N_i is the percentage observed over 50 chat rounds -- how was this chosen? Similarly, how were the choices for ensemble scale chosen?

How should we expect the runtime of the algorithm to scale to model in the 100B+ parameter range?

**Reasons To Accept:**

The setting of attacking one agent to compromise a Multi-Agent System is a compelling problem, and the authors formalize kth-order, N-spread, and one-intervention attacks in the paper. The evaluation is comprehensive and includes three scenarios, several open-source models, and ablations across several conditions such as suffix length, ensemble scale, etc.

**Reasons To Reject:**

I think a limitation of these analyses is the extent to which these attacks rely on direct message passing between agents. There are naturally settings where agents are asked to create summaries of previous steps that would mitigate these threats.

---

> ### Author Response · Authors · 2025-06-02
>
> Thanks for the time and the review.
>
> **Response to Weakness 1:  *Rely on direct message passing between agents.***
>
> If the agent is asked to summarize the previous questions, then we can add these cases also in the optimization process. Such as, we can add one loss term aiming at Input: ‘Summarize the previous information. Some context.. *Harmful content + suffix*’; Output: *Harmful content + suffix*. This summary scenario is a little bit similar to scenario 3, where the second agent is asked to rewrite the input. We can add these cases to the optimization ensemble cases.
>
> **Response to Question 1: *Statistical robustness checks.***
>
> Following previous works (GCG, ADC), we keep the LLM token sampling strategy as deterministic (greedy sampling), and the random seed as a fixed number. Based on our observation, the ASR isn’t really sensitive to the random seed, but the time for finding the adversarial suffix may vary.
>
> **Response to Question 2: *Hyperparameters in evaluation settings, how realistic is the evaluation?***
>
> **Rounds for N_i.** In the ideal case where once one infected agent talks to a non-infected agent, the non-infected agent gets infected, then the expected rounds for infecting all agents is 33662/3825 ≈ 8.80. However, since the suffixes are not that effective, and the randomly sampled order is not predefined, the average number of rounds for a successful attack will take around 30 rounds. So we set the number for N_i to 50.
>
> **Ensemble scale.** We did an ablation on the ensemble scale in tab.2, we find that even though a larger number of ensembles can lead to better performance, it kind of saturates. So 40 is a reasonable number for balancing the cost and the attack success rate.
>
> Overall, the evaluation settings are hard, with random order, random data sampled from Wikitext. The real-world settings for some multi-agent systems (MAS) now, is more similar to a predefined order. For example, ChatDev defines that the order is CEO agent first, then coder agent, UI designer, agent, and there will be multiple rounds between code review agent and coder/UI agent. Overall, the proposed evaluation setting is harder than some of the current multi-agent systems.
>
> **Response to Question 3: *Runtime for model of 100B+***
>
> Since the method can be transferred to black-box models such as GPT-o1, we suggest using transferability to attack larger models. On a larger model, the optimization time as well as the GPU memory will be costly.

---

### Official Review · Reviewer_78LB · 2025-06-02

**Rating:** 8
**Confidence:** 4
**Ethics Flag:** 1

**Summary:**

This paper proposes CODES, a new adversarial attack framework targeting LLM-based MAS through a single-shot suffix optimization. The authors define three realistic threat scenarios and demonstrate that CODES achieves high attack success rates with lower computational cost compared to prior methods. The work highlights critical vulnerabilities in MAS, including transferability to black-box models.

**Reasons To Accept:**

Overall, I enjoyed the idea, and it was interesting to see concrete empirical results across all three realistic attack scenarios.

1. The paper proposes three well-defined and realistic MAS attack scenarios, grounded in how real-world LLM-based multi-agent systems operate with limited user access.

2. The proposed CODES framework effectively combines continuous optimization and discrete search, leading to more stable and transferable adversarial suffixes compared to existing methods, with theoretical motivations grounded in loss landscape smoothness and propagation stability.

3. CODES achieves significantly higher ASR with lower computational cost, demonstrates transferability to black-box models (which I particularly liked), and includes thorough ablations on loss, ensemble scale, and generalization.

**Reasons To Reject:**

1. The paper does not offer concrete mitigation strategies or alignment improvements.

2. Relies solely on automated metrics; lacks analysis of the qualitative severity or perception of harmful outputs.

3. The MAS attack setting is only evaluated on small-scale, simulated agent groups (3–8 agents), limiting the claim that CODES poses a systemic threat to real-world, large-scale multi-agent systems.

(Apologies for the late review. I would appreciate it if the authors could provide a brief, high-level response regarding potential remedies.)

---

> ### Author Response · Authors · 2025-06-02
> **Responses to Reviewer 78LB**
>
> Thanks for the time and the review.
>
> **Response to Weakness 1: *Mitigation strategies.***
>
> For mitigating harmful content (scenarios 1 and 3), using a content filter can mitigate the harmfulness. However, a filter usually suffers from over-refusal issues, which can harm the performance of MAS.
>
> For mitigating stealthy behavior, such as saving a confidential file (scenario 2), training another agent for identifying unsafe code or behavior can be helpful.
>
> **Response to Weakness 2:  *Analysis of the qualitative severity.***
>
> The evaluation of qualitative severity of harmful content is hard and inaccurate [1,2]. Qualitative severity can be evaluated by human annotators, and we will add a small-scale human evaluation later using a survey.
> [1] Rauh, Maribeth, et al. "Gaps in the Safety Evaluation of Generative AI.”
> [2] Burden, John. "Evaluating ai evaluation: Perils and prospects."
>
>
> **Response to Weakness 3:  *Small-scale, simulated agent groups.***
>
> The development of AI agents is fast, currently, the agent groups usually contain fewer than 10 agents, such as ChatDev uses 7 agents for developing software, and AgentVerse has 9 agents for task-solving. There will be a larger scale of agents' communication in the future, say, if all the vehicles are connected within a transportation network, they will have a very sophisticated communication algorithm. We mainly investigate the random non-predefined order in this paper. We’ll keep this work updated and see how it performs in large-scale real-world settings.

---

### Decision · Program_Chairs · 2025-07-08

**Decision:**

Accept

**Comment:**

The paper introduces an adversarial attack framework targeting multi-agent systems through a single interaction with one of the agents. Aims to generate adversarial suffixes that transfer across agents. Some additional analyses find that the found attack patterns may even propagate to a black-box setup.

While the reviewers disagree significantly in their scores, there is consensus that the topic is timely and relevant and that the idea of combining continuous and discrete optimization is interesting. The paper includes a rigorous set of evaluations and multiple case studies.

I find that most drawbacks can be addressed within the scope of a camera-ready edits. Multiple reviewers find the paper dense and request improved clarity, some which which will be helped by the discussions provided throughout the rebuttal period. The reviews further sparked reviewers to ask about more real-world scenarios and experiments which I count as positive since it indicates that the paper led to interest.

Given that edits can improve clarity and discuss applicability of other SotA methods, I will recommend accept.